# Traumatic Experiences in Childhood and Maternal Depressive Symptomatology: Their Impact on Parenting in Preschool

**DOI:** 10.3390/children10010055

**Published:** 2022-12-27

**Authors:** María Pía Santelices, María de los Ángeles Fernández, Jaqueline Wendland

**Affiliations:** 1Centro de Investigación del Abuso y la Adversidad Temprana, CUIDA, Facultad de Ciencias Sociales, Pontificia Universidad Católica de Chile, Santiago 7810000, Chile; 2Escuela de Psicología, Pontificia Universidad Católica de Chile, Santiago 7810000, Chile; 3Laboratoire de Psychopathologie et Processus de Santé, Université Paris Cité, F-92100 Boulogne-Billancourt, France

**Keywords:** mother–child interaction, parenting, childhood trauma

## Abstract

The goal of this study is to describe and analyze the relationship between childhood trauma and depressive symptoms and its relation to the parental interactions of mothers (19–47 years) with their 3–4 year old preschool children. Parental interactions, traumatic experiences in mothers’ childhood, and current depressive symptoms were measured using the Parenting Interactions with Children: Checklist of Observations Linked to Outcomes (PICCOLO), the Childhood Trauma Questionnaire Short Form (CTQ), and the Beck Depression Inventory (BDI-I), respectively. A nonclinical sample of 81 Chilean mothers with their children was used. Results show that the presence of trauma in mothers’ childhood has an impact on parenting; specifically, mothers with an emotional neglect experience showed greater difficulties in adequately promoting autonomy in their children.

## 1. Introduction

### 1.1. Developmental Parenting

Parenting styles have been largely studied, as well as the factors that influence these styles and competences. Ref. [1] posed the Developmental Parenting Model as a type of childrearing that supports children’s development and growth. This model is based on the observation of parenting interactions, i.e., activities that promote the child’s social, cognitive, and linguistic development, which are grouped into affection, responsiveness or ability to respond, encouragement or stimulation, and teaching. These four dimensions contain core competences that all significant adults should deploy interacting with their children, although they have been mainly researched in mothers.

Parental competences can be positively or negatively affected by social and contextual factors. Parents’ educational level has been identified as a positive predictor in parenting [2], as education provides parents with relevant cognitive resources that help them become more effective in their parenting role [3]. For example, parents with more years of schooling reach higher levels of self-efficacy, which should make them feel more capable of managing the responsibility that parenting entails [4,5]. Studies have shown that lower socioeconomic status (SES) is associated with higher levels of perceived parental stress, a higher risk of tension accumulation, and a higher probability of parents’ abandonment of their parenting functions [6,7]. Furthermore, it has been observed that mothers who have suffered trauma in their infancy tend to react more aggressively to stress in the child by responding inadequately to their needs [8]. In addition, mothers with depressive symptoms are more likely to be aggressive, insensitive, or emotionally unresponsive, and they can have a tendency to be negligent [9]. In this regard, depression is believed to lead to psychosocial risks to the child. Maternal depression can affect the way the child will self-regulate in the future [10], as the mother’s depression and the child’s regulation difficulties may be connected through observational learning when the child is exposed to the mother’s depressive symptoms. Despite this, mothers’ capabilities can counter the negative consequences of trauma and mitigate depressive symptomatology [11].

### 1.2. Maternal Childhood Trauma

In this study, we regard trauma as any type of maltreatment experienced by the mother during childhood. Trauma is configured by any episode of sexual, emotional, and/or physical abuse, and/or by any form of emotional and/or physical neglect included in the trauma model advanced by [12]. This model regards sexual abuse as any sexual contact or behavior between a child under 18 years of age and a person over 18 years of age, physical abuse as bodily aggressions by an adult over 18 years of age against a child, emotional abuse as verbal aggressions against a child’s self-worth and wellbeing or any other humiliating or degrading behavior directed to a child by an adult, physical neglect as the caregivers’ failure to satisfy the child’s basic physical needs, including, food, shelter, clothing, safety, and healthcare, along with poor parental supervision if the child’s safety is not ensured, and emotional neglect as the caregivers’ failure to satisfy the child’s emotional and psychological needs, including love, a sense of belonging, care, and support [12].

Empirical studies have shown that parents who have experienced trauma in childhood could fail to recognize the meaning of the child’s signals of need and tend to engage in punitive or negligent practices due to interpreting the child’s biological needs as negative, responding with anger and rejection [13]. Furthermore, they tend to show low sensitivity, lack of enjoyment, intrusive behaviors, negative emotions, and punitive behaviors during interaction and a tendency to negatively perceive the behavior of their babies [14]. On the other hand, emotional abuse in mothers’ childhood is associated with a decrease in their acceptance of the child and an increase in their psychological control, all of which affect children’s psychological and emotional development and curtail their thinking skills and self-expression [15]. Research has shown that mothers who have suffered childhood trauma linked to emotional abuse lack suitable models of warm and consistent parenting, tending to react emotionally rather than with a balance of emotion and reason [16].

Other associated risk factors in parents who maltreat their children, either through abuse or neglect, are the presence of depressive symptomatology, impulse control difficulties [17,18], and psychiatric comorbidities [19]. Early childhood risk factors leading to major depressive symptomatology in adulthood include maltreatment and trauma (depression, low self-esteem, psycho-physiological reactions, and coercive discipline) [20]. This observation is supported by several studies that showed relevant associations between childhood abuse and depressive symptomatology in adulthood, along with suboptimal childrearing patterns [11,21].

### 1.3. Depression and Parenting

Strong evidence supports the view that depressive symptoms alter healthy parental practices [22,23]. The scientific literature has shown that mothers who display depressive symptomatology may also experience problems in their maternal role: diminished emotionality, deteriorated communication, and an increase in hostility and resentment [24]. Evidence suggests that mothers with depressive symptomatology have difficulties providing their children with the necessary resources for their development, are plagued by thoughts about harming their children [25], have lower levels of maternal sensitivity [14], find it difficult to hold conversations with their children that encourage the identification and the adequate expression of emotions [26], and have an impaired ability to perceive the child’s needs and provide emotionally nurturing care [27]. In addition, it has been demonstrated that parents with severe depressive symptoms blame their children by ascribing negative intentions to them [28], which makes them increase punitive actions.

### 1.4. Current Study

It is of great importance to study parental interactions with children, as well as the factors that influence these interactions, since early childhood is a key stage for socioemotional development, and it has been shown that parenting behaviors have deep effects on children’s emotional functioning [29]. Research on parental competences and the factors that determine them has tended to focus on early childhood (0 to 3 years of age) rather than on later periods such as the preschool stage. In addition, few studies have examined the specific consequences of each type of trauma on parenting, together with factors such as the presence of depressive symptomatology or low education levels. The main objective of the present study is to analyze the relationship between traumatic experiences in Chilean mothers’ childhood and the parental competences of mothers with children who attend preschool, considering the presence or absence of depressive symptomatology and maternal education. The study is expected to reveal reduced parental competences in the presence of childhood trauma, maternal depressive symptomatology, and/or a low maternal education level.

## 2. Method

### 2.1. Design

The methodology used was quantitative, with a descriptive, nonexperimental, and cross-sectional design. The maternal variables of interest were described on the basis of self-report questionnaires and the observation of mother–child interactions by filming them for 5 min in a semi-directed playful interaction.

### 2.2. Participants

The study included 81 mothers and their 3–4 year old children who attended seven preschools belonging to the National Preschool Association, located in the Metropolitan Region and serving a mid- to low-SES population. Only mothers who were their children’s main caregivers were included.

Sample size was calculated using the G power program for complex statistical analysis (in this case, linear regressions). It was calculated a priori, hoping to obtain a statistical power of 0.80. A median effect size f2 = 0.15 was considered, with a significance level of α = 0.05, obtaining a necessary sample size of 92 people to reach a statistical power of 0.80. The significance level was established according to the convention in the field of psychology research to set the confidence level of the estimate at 0.05 [30]. Given that the sample reached was 81 people, the statistical power was recalculated, obtaining a result of 0.74, approaching the expected power.

### 2.3. Procedure and Analysis

Participants were recruited through preschools of the National Preschool Association. After contacting seven preschools and presenting the project to them, parents of the upper intermediate level (3–4 year old children) were invited to participate voluntarily. The mothers signed an informed consent letter, while the children gave their verbal assent to participate. The assessment procedure was conducted in the preschools. Administration and coding were carried out by raters previously trained by experts. The Scientific Ethics Committee of the institution that developed this research approved this study (approval code 1130786-2013).

In the first place, the observers were certified as PICCOLLO coders with an individual reliable score; then, they calibrated amongst themselves until they reached the desired score to be able to start codifying the videos of this study. The data analysis stage first involved descriptive results of frequency of the variables of interest, followed by a correlation analysis to evaluate the degree of association between the variables of interest in the adult (presence of childhood trauma and maternal depressive symptomatology), while controlling some variables such as the education level. Lastly, multiple regression models were used to evaluate the impact of childhood trauma and depressive symptomatology on the parental competences observed (affection, responsiveness, encouragement, and teaching), considering each of the subitems of the scales employed. The analysis was conducted using SPSS v21.

### 2.4. Instruments

Parenting Interactions with Children: Checklist of Observations Linked to Outcomes (PICCOLO) (for interaction between mother and child) [1]. This instrument measures positive parenting through the observation of a mother–child routine interaction and was developed to encourage behaviors in parents that support children’s early development. It contains 29 items that measure parental development, according to the theory of early child development. These items are grouped into four domains: affection, responsiveness, encouragement, and teaching. The instrument provides subtotals for each dimension with a maximum of 68 points. Specific cutoff scores for each dimension were obtained from [1]. The average ranges of scores are 9–12 points for the affection dimension, 10–13 points for responsiveness, 8–12 points for encouragement, and 6–11 points for teaching. The original instrument has a reliability index of 0.78. In the present study, two raters were required to achieve a kappa index of 0.79. Cronbach’s alphas for the affection, responsiveness, encouragement, and teaching domains reached values of 0.56, 0.72, 0.62, and 0.72 respectively. Due to the low reliability of the affection dimension, it was not included in the current analysis. The videos were codified by two qualified observers; before starting to watch the videos and score each item, they were calibrated to reach the desired inter-reliability score.

Beck Depression Inventory (BDI-I) (mothers). The BDI is an instrument developed by Beck [31] that has been extensively used in multiple studies due to its straightforward administration and tabulation. This instrument assesses the presence of depressive symptomatology in adults and adolescents aged 13 years and up through a self-response questionnaire that comprises 21 Likert-type items that describe the most frequent clinical symptoms of depression sufferers, such as sadness, crying, loss of pleasure, feelings of failure and guilt, suicidal thoughts or wishes, and pessimism. Each item is made up of four statements about the intensity of the symptom, ranging from 0 (absent or mild) to 3 (very intense). If the person selects several response categories in an item, the category with the highest score is considered. The instrument yields an overall score ranging from 0 (minimum) to 63 points (maximum), with higher scores representing a more extensive presence of depressive symptomatology and, consequently, more risk. Cutoff scores were established in order to group respondents into the following categories: 0–13, minimum depression; 14–19, mild depression; 20–28, moderate depression; 29–63, severe depression. This instrument has shown a high degree of internal consistency in clinical and nonclinical samples, with an alpha coefficient of approximately 0.82 [31].

Childhood Trauma Questionnaire Short Form (CTQ) (mothers). The abbreviated CTQ is an instrument developed by [32] which assesses the presence of traumatic events in childhood. It is short, easy to administer, and relatively noninvasive. This widely used scale comprises 28 Likert-type items that can be completed in 5–10 min and can be administered from 12 years of age onward. It includes five subscales that make it possible to identify traumatic childhood experiences involving emotional, physical, and sexual abuse, as well as emotional and physical neglect. Each answer belongs to one or more subscales and can be assigned a score between 1 and 5. Cutoff points are defined to determine the type of maltreatment experienced: under 5, no trauma; over 6, sexual abuse; over 8, physical abuse and neglect; over 12, emotional abuse; over 13, emotional neglect. This instrument has been shown to have good psychometric properties, adequately fitting the five-factor structure in clinical and nonclinical samples It has an internal consistency of 0.95 and an alpha coefficient of 0.9 [32].

Sociodemographic questionnaire (mothers). The sociodemographic questionnaire used in this study was developed by the research team in order to collect information to characterize the sample. It included family-related and individual aspects of the preschooler included in the sample, along with information about the socioeconomic and educational level of the participating adult.

## 3. Results

### 3.1. Descriptive Results

Mothers were 29 years old on average (SD: 6.4, ranging from 19 to 47 years); 21% of them had not completed their school studies, 69% had finished school (secondary or technical education), and 6% had a university education. As for the preschoolers, the sample was made up of 40 girls (49%) and 41 boys (51%) who were 43 months old on average (SD = 2.7, ranging from 36 to 47 months). The inclusion criteria were mothers older than 18 years old with children aged 36 to 47 months. Exclusion criteria were women who were not mothers or mothers with children younger than 36 months or older than 47 months.

It Is to be noted that the original PICCOLO instrument has 29 items that are grouped into four domains: affection, responsiveness, encouragement, and teaching. Due to the low reliability of the affection dimension, it was not included in the current analysis.

Table 1 shows the distribution of the quality of the mothers’ parental interactions in each of their dimensions. It can be observed that 93% of the mothers engaged in average or above-average teaching interactions, a value that reached 76% for responsiveness and 74% for encouragement.

Table 2 reveals the presence of childhood trauma in the participating mothers. Overall, 56% of mothers reported having experienced one or more traumas in childhood, while 33% stated that they had suffered more than one type of trauma in childhood.

Regarding the presence of depressive symptomatology, 81% of the sample displayed a minimum level of symptomatology, and 19% showed mild or moderate symptoms. Table 3 shows the frequency of comorbidity between reported trauma in the participating mothers’ childhood and their current depressive symptomatology.

### 3.2. Correlation Analyses

Bivariate Pearson correlation was performed among the mothers’ educational level, parental interactions, reported childhood trauma, and maternal depressive symptomatology. The correlation matrix presented in Table 4 reveals significant associations of the mothers’ education level and the presence of physical neglect with the teaching domain. For this reason, the following analyses were conducted, controlling for the mothers’ education level.

As shown in Table 5, a partial correlation was performed, controlling for the mothers’ education level, between the quality of parental interactions and reported trauma in the mothers’ childhood. The correlation matrix revealed negative and significant associations between the encouragement domain and the presence of emotional neglect in the mothers’ childhood (*r* = −0.356, *p* = 0.001). This suggests that, in the case of mothers who suffered emotional neglect as children, the quality of parental interactions was lower in the encouragement domain.

### 3.3. Regression Analyses

Next, regression models were generated using the encouragement dimension as the dependent variable and the multiple types of trauma as independent variables. The results, as reported in Table 6, show that the presence of emotional neglect in the mothers’ childhood was the strongest predictor of reduced parental interaction quality in the encouragement domain (*β* = −0.557). Regression models were also generated for the responsiveness and teaching domains; however, the results obtained were nonsignificant.

## 4. Conclusions and Discussion

The present study analyzed parental interactions in connection to the presence of maternal childhood trauma and depressive symptomatology. In addition, the study analyzed whether mothers’ education level had an impact on the variables studied, considering the evidence for the influence of this variable on parenting [2,5]. The study was mainly focused on parental interactions and maternal childhood trauma because this is a relatively unexplored dimension in the Chilean context. This aspect is relevant due to the high prevalence of child maltreatment and depressive symptomatology in women of childbearing age in Chile [33].

It was observed that 19% of the mothers displayed depressive symptomatology, a significantly lower rate than the national average of 27.9% for the female population in Chile. The sample appears not to behave in the same way as the mid- to low-SES Chilean population. Regarding the quality of parental interactions, a comparison with the US original sample revealed that the Chilean sample displayed better teaching interactions; however, this was not true of the encouragement dimension, as the Chilean mothers in the sample studied behaved less encouragingly with their children. In the case of responsive interactions, quality remained stable.

The results obtained indicate that the presence of trauma in mothers’ childhood has a strong impact on their future parenting. Mothers who have experienced emotional and psychological deficits in childhood due to their caregivers’ emotional negligence in terms of providing love, a sense of belonging, care, and support will find it more difficult to properly foster their children’s autonomy, interests, decision-making skills, motivation to tackle new challenges, and cognitive, social, and linguistic development. The stimulation of children’s interests and their ability to decide will be lower in mothers who have experienced emotional neglect in childhood, which is consistent with the existing literature [34]. Emotionally neglected mothers will have difficulty being emotionally available, which usually leads to their failure to value children’s efforts [35]. In addition, they tend to interact with them in a hostile way [36]. Furthermore, the presence of emotional neglect in childhood is positively correlated with lower levels of maternal sensitivity [37]. 

If a mother’s parents or caregivers had difficulties meeting her emotional and psychological needs (in other words, if they failed to provide her with a psychological space), she will be less likely to encourage her child’s autonomy. In view of this, these mothers’ daughters may find it hard to replicate as mothers that which they never experienced in childhood, as suggested by the transgenerational cycle of abuse. Therefore, when these girls become mothers, they will not have internalized the interaction models that would have allowed them to validate their children considering their needs, interests, and wishes, which is indicative of low levels of mother–child affective synchrony. Unmet emotional and psychological needs in childhood lead to a reduction in mothers’ affective and reciprocal behaviors, such as changing the rhythm of an activity or shifting to a different one altogether depending on the child’s interests or needs, following what they are doing, responding to the emotions expressed by the child, and looking at or responding to them when they speak or utters a sound.

In the present study, reported trauma in the mother’s childhood was not found to be linked to more depressive symptomatology. This finding is not consistent with the literature [9,17,22,38], but there is evidence that resilience can significantly counter trauma, even mitigating depressive symptoms [39]. Nevertheless, this weak association may be due to the low prevalence of depressive symptomatology observed in the sample and its low comorbidity. Therefore, the lack of a connection between these conditions should not be regarded as a significant factor for understanding trauma.

When trauma was reported, specifically sexual abuse in the mother’s childhood, the quality of parental interactions was not found to be lower. This is not consistent with most of the literature [11,40], although some researchers did not observe this association [11,41]. It is important to state that sexual abuse may have occurred outside of the nuclear family, and the family may have intervened in an adequate and protective manner. On the other hand, the joint presence of childhood trauma and depressive symptomatology in the mother may not fully account for the reduction in parental competences. This is not in line with the literature, which has identified strong associations between a history of childhood abuse and depressive symptomatology in adulthood, along with suboptimal childrearing patterns [11].

The present study showed that the presence of depressive symptomatology in this nonclinical sample of mothers did not significantly impact parental interactions, and that trauma can be singled out as the main culprit of their deterioration. In this regard, trauma characterized by emotional abandonment by mothers’ significant figures in childhood is believed to be the best predictor of a reduction in parental interaction quality. This is especially important given the predictive weight ascribed to traumatic experiences in early childhood with respect to the victim’s future parenting [15]. Nevertheless, the associations found may be conditioned by the limited presence of depressive symptomatology in the sample.

The non-incidence of depressive symptomatology on parental interactions may be explained by the fact that the sample used was nonclinical and displayed a minimum or mild degree of depressive symptoms. Likewise, due to the voluntary nature of these studies, they require some degree of psychic energy to deal with the assessments involved; therefore, the participating mothers may display less symptoms and resistance. Considering this, one of the limitations of the present study was the lack of information about whether the participants had previously received psychotherapeutic care (which would have allowed them to work through their trauma or depressive symptomatology) and whether they were taking some type of medication. In addition, the limitations of self-report questionnaires must also be mentioned. The homogeneity of the sample in terms of SES and education levels (middle to lower class), as well as its size, constitutes another limitation. Lastly, another potential limitation may be related to the low Cronbach’s alpha values calculated for the encouragement domain and the fact that the PICCOLO and CTQ have not been validated for Chile.

This study is relevant since it opens the discussion on the importance of mental health factors that influence parenting in early childhood and, therefore, the optimal emotional development of children. Future studies could consider additional factors such as parental stress, which could have a major influence on parental competences due to parents’ physical or emotional fatigue. This may affect their attitude and the time that they devote to their children, having a strong impact on the quality of play. The main clinical and research implications derived from this study are the need to examine the mother’s family history, as well as her prior and/or current treatments, when the child is the reason for seeking help, to reduce mothers’ stigmatization as bad caregivers and promote empathy with them if they have a history of childhood neglect. Future research could include child-related variables, considering children’s reactions and including psychoeducational or feedback interventions, as well as subsequent measures intended to follow up the progress made by mothers and their children. Another potential line of research could involve generating comparative models of interactions between fathers and mothers if one of them has experienced maltreatment in childhood.

## Figures and Tables

**Table 1 children-10-00055-t001:** Distribution of the mothers’ parental interaction quality.

	Responsiveness	Encouragement	Teaching
Weakness	24%	26%	7%
Average	49%	68%	61%
Strength	27%	6%	32%

**Table 2 children-10-00055-t002:** Frequency of reported trauma in the mothers’ childhood.

	N	%
No trauma	36	44
1 type of trauma	19	23
2 types of trauma	11	14
3 types of trauma	4	5
4 types of trauma	8	10
5 types of trauma	3	4
Total	81	100

**Table 3 children-10-00055-t003:** Frequency of comorbidity between childhood trauma and depressive symptomatology.

Type of Trauma		Depressive Symptomatology				Total	
		Absence		Presence			
		N	%	N	%	N	%
Physical abuse	Absence	51	63	11	14	81	100
	Presence	15	19	4	5		
Sexual abuse	Absence	54	67	13	16	81	100
	Presence	12	15	2	2		
Emotional abuse	Absence	51	63	8	10	81	100
	Presence	15	19	7	9		
Physical neglect	Absence	40	49	11	14	81	100
	Presence	26	32	4	5		
Emotional neglect	Absence	54	67	13	16	81	100
	Presence	12	15	2	2		

**Table 4 children-10-00055-t004:** Correlations between the mothers’ education level and the reported childhood trauma and depressive symptomatology variables.

	Mother’s Education Level	
	*r*	*p*
Depressive symptomatology	−0.044	0.702
Physical abuse	−0.019	0.870
Sexual abuse	−0.117	0.305
Emotional abuse	−0.102	0.369
Physical neglect	−0.247 *	0.028
Emotional neglect	−0.181	0.111
Responsiveness	0.172	0.130
Encouragement	0.193	0.089
Teaching	0.251 *	0.026

* *p* ≤ 0.05.

**Table 5 children-10-00055-t005:** Partial correlation controlling for the mothers’ education level.

		Physical Abuse	Sexual Abuse	Emotional Abuse	Physical Neglect	Emotional Neglect	Depressive Symptoms
Responsiveness	r	−0.019	0.133	0.08	−0.122	−0.121	−0.036
	*p*	0.867	0.247	0.488	0.289	0.293	0.758
Encouragement	r	0	0.018	−0.061	−0.2	−0.356 *	−0.162
	*p*	0.998	0.876	0.598	0.079	0.001	0.156
Teaching	r	−0.078	0.056	0.038	−0.088	−0.065	−0.092
	*p*	0.495	0.627	0.743	0.445	0.571	0.422

* *p* ≤ 0.05.

**Table 6 children-10-00055-t006:** Linear regression models using the encouragement domain as dependent variable.

	1	2	3	4	5	6	7
Mother’s education level	0.196	0.149	0.252	0.100	0.064	0.118	0.137
Physical abuse	−0.012						0.002
Sexual abuse		0.023					0.224
Emotional abuse			−0.049				0.206
Physical neglect				−0.195			−0.051
Emotional neglect					−0.351 **		−0.557 **
Dependent variable: encouragement							

** *p* ≤ 0.001.

## Data Availability

The data presented in this study are available on request from the corresponding author.

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
