# Peer review of "Traumatic Experiences in Childhood and Maternal Depressive Symptomatology: Their Impact on Parenting in Preschool"

_children, 2022, doi:10.3390/children10010055_

Round 1

Reviewer 1 Report

Thanks for your effort working on this paper.
This study is significant in that it provides empirical evidence for the relationship between mothers' childhood traumatic experiences and mothers' parental interactions. However, a minor revision is required for the following content.
The evidence for mothers of preschool children is not sufficiently presented. It is hoped that the importance of the parental role at this time, including the developmental characteristics of preschool children, should be emphasized.
Please insert the approval number of the Research Ethics Committee.
Research tool PICCOLO is designed to be written through observation. Please describe in detail how the researcher worked to increase inter-observer reliability.
Highlight the strengths of your findings and add them to the discussion
References: Revision old references to recent references as much as possible.

Author Response

Dear reviewer,

Thank you for your comments. We have reviewed the article and included your suggestions. We're open to making any other changes needed. 

Kind regards

The evidence for mothers of preschool children is not sufficiently presented. It is hoped that the importance of the parental role at this time, including the developmental characteristics of preschool children, should be emphasized.

More evidence was added

○ Please insert the approval number of the Research Ethics Committee.

Addressed

○ Research tool PICCOLO is designed to be written through observation. Please describe in detail how the researcher worked to increase inter-observer reliability.

Addressed 

○ Highlight the strengths of your findings and add them to the discussion

Addressed

○ References: Revision old references to recent references as much as possible.

They have been reviewed

Reviewer 2 Report

Introduction:

The introduction is written like a literature review section. The first sentence of the introduction looks like a section of a literature review. The introduction can be re-written in simple language keeping in view the research objectives. Moreover, the introduction is quite lengthy and can be shortened.  

Methods:

The first sentence lines #130-132 are not understandable. Rephrase these sentences and explicitly mention the research design used for this study.

 Line#138-142 should be mentioned in the result section. 

Sample size calculation details are missing.

In the abstract section, it is mentioned that "mothers (19-47 11 years) with their preschool children (36-47 months)". However, in the method section, lines 144-145, it is mentioned that "parents of the upper intermediate level (3 to 4-year-old children) were invited to participate voluntarily". I could not understand the exact population of the study. Moreover, it is also mentioned that the authors received verbal assent from children. In addition, the study participant section does not report that the children were also part of this study. It is only mentioned that mothers were part of this study. Moreover, the assent procedure is not mentioned in the method section. 

The participant inclusion and exclusion criteria are not mentioned.

The interview process is not written clearly. 

The analysis plan is not much clear. For example, how the coding was done for analysis. 

The authors mentioned questionnaires in the methods section, but that is not clear which questionnaire/s was/were used for children. 

Results:

The child data is not reflected in the result section. 

The conclusions can be written separately from the discussion section.

Author Response

Dear reviewer,

Thank you for your comments. We have reviewed the article and included your suggestions. We're open to making any other changes needed. 

Kind regards

Introduction:

The introduction is written like a literature review section. The first sentence of the introduction looks like a section of a literature review. The introduction can be re-written in simple language keeping in view the research objectives. Moreover, the introduction is quite lengthy and can be shortened.  

The introduction has been rewritten. 

Methods:

The first sentence lines #130-132 are not understandable. Rephrase these sentences and explicitly mention the research design used for this study.

Rephrased.

 Line#138-142 should be mentioned in the result section. 

Addresed.

Sample size calculation details are missing.

Addressed.

In the abstract section, it is mentioned that "mothers (19-47 11 years) with their preschool children (36-47 months)". However, in the method section, lines 144-145, it is mentioned that "parents of the upper intermediate level (3 to 4-year-old children) were invited to participate voluntarily". I could not understand the exact population of the study. Moreover, it is also mentioned that the authors received verbal assent from children. In addition, the study participant section does not report that the children were also part of this study. It is only mentioned that mothers were part of this study. Moreover, the assent procedure is not mentioned in the method section. 

Changes have been made so that the method section is clearer.

The participant inclusion and exclusion criteria are not mentioned.

Addressed.

The interview process is not written clearly. 

Addressed.

The analysis plan is not much clear. For example, how the coding was done for analysis. 

Addressed.

The authors mentioned questionnaires in the methods section, but that is not clear which questionnaire/s was/were used for children. 

This has been clarified.

Results:

The child data is not reflected in the result section. 

The only instrument where children were considered was the PICCOLO. This instrument gives results on the mother-child interaction, but there are no child data on their own.

The conclusions can be written separately from the discussion section.

Addressed.

Round 2

Reviewer 2 Report

Some previous instructions did not follow by the reviewers in the revised draft

Methods:

Move Lines #110-115 in the result section.

Sample size calculation detail still missing 

The details on how the participants were recruited are missing

I think the "conclusions" heading should be "discussion" and "Discussion" should be the "conclusions"

Author Response

Move Lines #110-115 in the result section.

- They have been moved

Sample size calculation detail still missing 

- It has now been added: Sample size was calculated using the G power program for complex statistical analysis, in this case, linear regressions. It was calculated a priori, hoping to obtain a statistical power of 0.80. A median effect size f2 = 0.15 was considered, with a significance level of α = .05, obtaining a necessary sample size of 92 people to reach a statistical power of 0.80. The significance level was established based on the convention in the field of psychology research to set the confidence level of the estimate at .05 (Castro & Martini, 2014). Given that the sample reached was 81 people, the statistical power was recalculated, obtaining a result of 0.74, approaching the expected power.

The details on how the participants were recruited are missing: Addressed. Participants were recruited through preschools of the National Preschool Association. After contacting 7 preschools and presenting the project to them, parents of the upper intermediate level (3 to 4-year-old children) were invited to participate voluntarily. 

I think the "conclusions" heading should be "discussion" and "Discussion" should be the "conclusions": All the team believes its better to have one section. It has been modified. 
